

# Differential nest-defense to perceived danger in urban and rural areas by female Eurasian sparrowhawk (*Accipiter nisus*)

Tomas Kunca[1] and Reuven Yosef[2]

[1] Faculty of Environmental Sciences, Czech University of Life Sciences, Prague, Czech Republic
[2] Eilat Campus, Ben-Gurion University of the Negev, Eilat, Israel

## ABSTRACT

The reaction of wildlife to humans is known to differ with surroundings. In urban environments that provide suitable habitats for breeding birds, animals adapt to humans and their response is accordingly altered. This study examined the nest defense behavior of female Eurasian sparrowhawks (*Accipiter nisus*) during the breeding season in urban and rural areas of Prague. The females showed four different types of reaction to humans that approached the nest and differed significantly between the two study areas. Contrary to expectations, urban nesting females were more aggressive than rural conspecifics. The intensity of response increased as the season progressed, and females defended their broods to a much greater degree than their clutches in both urban and rural habitats, suggesting a differential effort as a function of their relative investment in the breeding attempt conforming with the parental investment hypothesis.

## INTRODUCTION

Wildlife are known react to perceived danger, including human presence, which affects their normal behavior and responses in a wide range of species-specific ways (*Dukas*, *2001*; *Wojciechowski & Yosef*, *2011*). The animals reaction to perceived danger is known to influence its fitness and life history, and to affect other behaviors such as inter- and intra-specific interactions and foraging considerations (e.g., *Grubb & Greenwald*, *1982*; *Morse*, *1986*; *Nonacs & Dill*, *1990*; *Watts*, *1990*; *Stamps & Bowers*, *1991*; *Slotow*, *1996*). *Whittaker & Knight* (*1998*) considered the reaction of an individual as the combination of learning and genetics.

Expanding human populations in almost every habitat on the globe has resulted in wildlife having to contend with greater disturbances in their natural environments or outright destruction of existing habitats. The outcome of ever-increasing human-animal encounters influences the learning components of animals (*Knight & Temple*, *1986*). With the growing human population natural environments have become increasingly modified by humans and some species have reacted to such changes by habituation. Urban areas,

Corresponding author
Tomas Kunca, tkunca@gmail.com

in the form of cities, provide habitats for many animal species that willingly overcome ecological barriers of urbanization and adapt to the human-dictated conditions (*Luniak*, *2004*). Interest in urbanization of birds is not novel (*Erz*, *1966*) and the number of studies that describe avian responses to urbanization is immense and growing (*Marzluff, Bowman & Donelly*, *2001*). In particular, some raptor species find urban habitats very suitable as they provide large amounts of food in the form of human commensals (e.g., Feral Pigeons, *Columba livia*; Sparrows, *Passer spp.*; Brown Rats, *Rattus norvegicus*), quality nesting places and are free from some ecological pressures, such as persecution (*Chace & Walsh*, *2006*). However, in the presence of persecution, or high levels of disturbance resulting in stress or unsuccessful breeding, it would be maladaptive to habituate to human predominant environments and birds can be expected to display avoidance behavior or increased aggression towards the instigator of the disturbance (*Shannon et al.*, *2014*). Persecuted avian species are known to behave differently in rural and urban environments (*Kenney & Knight*, *1992*). As a result of direct persecution/disturbance and subsequent evasion, breeding birds react to perceived danger with stereotypic antipredator behavior. Nest predation is an important factor limiting breeding success and various birds view humans as potential predators (*Fisher et al.*, *2004*). Thus, defending the nest can reduce the parents' wasted energetic investment caused by the loss of a clutch or brood (*Grim*, *2008*). Studies of avian nest defense show an increase in the intensity of the parents' defense as the breeding period progresses, i.e., the more advanced the breeding stage, the greater the parents' energetic investment in the reproductive attempt, resulting in an increased effort to defend their young (e.g., *Merritt*, *1984*; *Shields*, *1984*; *Sergio & Bogliani*, *2001*). The aforementioned studies conform to the parental investment hypothesis (*Trivers*, *1972*; *Barash*, *1975*). However, *Knight & Temple* (*1986*) discovered that nest defense behavior was gradually modified by repeated visits to the nests, consequently resulting in the parents' loss of fear.

The Eurasian sparrowhawk (*Accipiter nisus*) is a common raptor that prefers to breed in woods and forests where it can find an abundance of prey (*Newton*, *1986*). Sparrowhawks started to breed in Prague in the early 1980's (*Šťastný, Randík & Hudec*, *1987*), between 1985–2004 the numbers of breeding pairs varied greatly (42–91 pairs), and eventually stabilized at 45–55 breeding pairs. However, because not all the individuals in the urban and surrounding rural areas are ringed for individual identification, we remain ignorant about the origins of the breeding population (sensu latu—the pioneer pairs; *Rutz*, *2008*), the turnover within the urban population, or if the urban population is a source or a sink. At present, ringing of the city-bred young show that they comprise a substantial proportion of the individuals that are recruited into the breeding urban population (Peške in *Fuchs et al.*, *2002*). In rural areas, sparrowhawks have been persecuted by gamekeepers for centuries and whilst it was largely stopped in the 1980's, sparrowhawks were still hunted in the Czech Republic in the 1990's (Myslivecká statistika *Ministerstvo Zemědělství*, *2014*).

In order to understand the influence of human activities on rural and urban sparrowhawk breeding pairs, we compared the reactions of breeding females to perceived danger. We hypothesized that we would find a greater degree of disparity between the behaviors of the urban and rural breeding females. We tested our hypothesis only on females due to the fact that, in sparrowhawks in the immediate vicinity of the nest, it is the female

sparrowhawks who usually repel intruders or react to perceived danger (*Newton*, *1986*). Hence, in order to check our hypotheses, we observed the nest defense of adult female sparrowhawks during the breeding season in an urban area with high human density where we assumed sparrowhawks to be habituated to human beings, and in a rural area in which sparrowhawks live in the wild and have minimal contact with humans.

## MATERIAL AND METHODS

### Study area

The urban study was conducted in the center of Prague, Czech Republic ($\sim$240 km$^2$, radius of 8.7 km around the epicenter of the city, N50° 4.53497′, E14°26.05478′). The area encompassed by the breeding sparrowhawks, included neighborhoods with residential housing, business districts and industrial areas. Sparrowhawks bred in parks, gardens and cemeteries. The rural study area was situated in the Liberec region of north Bohemia (N50°49.57548′, E14°35.28330′) where sparrowhawks are found in large tracks of natural forest. The distance between the two areas is $\sim$100 km. In both study areas, data were compiled simultaneously during the breeding season which extended from early May to June 2013. In our analysis, we included only those nests which were located during the building stage and prior to egg laying. In addition, two nests that failed to fledge young were not included in our analysis. The nest sites were only visited twice during the breeding cycle in order to avoid habituation of birds to human approach since multiple visits are known to influence the female's reaction (*Knight & Temple*, *1986*). At each of these visits, we conducted this experiment and also verified the breeding reproductive stage.

Eurasian sparrowhawk is characterized by a strong sexual dimorphism. The females are almost twice the size of the males and their undersides have grayish-brown striations, while those of the male are rusty in color. Another important fact is that the males very rarely incubate the eggs and do so only when the female is feeding, furthermore they never brood the young (*Newton*, *1986*). The higher pitch call of the male also helps distinguish between the two sexes. These differences allowed us to be confident in knowing which bird we were observing during our visits.

### Data collection

We recorded the response of each incubating female twice during the breeding season. Based on our data from previous years and on the different related behavior, we were able to estimate the different stages of the reproductive cycle. The first trial was conducted during the second half of incubation and the second trial during the first week of the nestling stage. We approached the nest from the nearest path most frequented by humans, in a very obvious manner, and once at the base of the nest tree, recorded the female's reactions for five minutes. To minimize any influence of inclement weather, nest visits were made during windless days with no precipitation. We categorized the female reactions into four behavioral responses. None of the observed females had reactions that spanned multiple behavioral responses:

(1) skittish: leaves the nest at our approach, remains silent throughout the visit and does not visit the nest the 5 min period during which we stand at the base of the tree;

(2)  alert: alert to our approach but leaves the nest only when we are at the base of the nest tree, utters warning calls, irregularly spotted passing the nest tree during the 5 min period;

(3)  angry: alert to our approach and does not leave the nest while we stand at the base of the nest tree, leaves the nest only after gentle shaking of the nest tree, utters repeated warning calls, remains in eye-contact with the nest from adjacent trees;

(4)  intense: alert to our approach, does not leave the nest while we stand at base of the nest tree, refuses to leave the nest even when the tree is shaken, responds by aggressive wing beating against nest edge, aggressive posturing/mantling on the nest.

We used a Nikon Forestry PRO (Nikon Vision, Tokyo, Japan) to measure nest height and distance to the path. We categorized the habitat, where the nest was located, into four types according to the number of trees and basal area within a radius of 100 m around the nest tree:

(1)  Very dense stand comprised of either young spruce (*Picea* spp.) or pine (*Pinus* spp.) with >15 cm diameter at breast height (DBH);

(2)  Dense stand comprised of either one tree species or mixed coniferous (spruce, pine and larch, *Larix* spp.) with diameter >30 cm DBH;

(3)  full-grown mixed forest with diameter <30 cm DBH;

(4)  Solitary, fully-grown trees in city parks, and cemeteries.

## Data analyses

Generalized linear models (GLM) with binomial distribution of errors of response variables (i.e., presence or absence of certain type of reaction) were employed to analyse possible effects of explanatory variables (locality—urban/rural; breeding stage—incubation/brooding; habitat; distance to path; individuality) using R (*R Development Core Team*, *2011*). Data on nest height and its distance from the path were logarithmically transformed to approach normality. The nest height (continuous variable) and habitat (categorical variable) were not independent ($F = 34.182$, $Df = 3$, $P < 0.001$) and therefore only the habitat was used in the analysis. Although this variable is more complex, it is more relevant to the object of our study. Individuality of the female was factored with a random effect. Individuality was used because each female's response was included in the model twice (incubation/brood). By use of the individuality index we avoided pseudoreplication given that two observations were conducted at each nest. We ran a model for each female's reaction resulting in four separate models. Full models, containing all explanatory variables, were then simplified, i.e., all non-significant explanatory variables ($P > 0.05$) were excluded step-by-step, using the backward selection procedure (*Crawley*, *2007*). Because of the nature of the statistical analysis, the significance level for the final results was tightened ($P < 0.01$) to reduce type I error. Before we used the final model we tested its parsimony and compared the null model with the full model. The full model did not improve the significance and the AIC value was low (null model AIC: 73.811, full model AIC: 62.907, used model AIC: 62.666). By running four different models, we were able to see the separate effect of the factors on each of the female's responses (cf. *Slamova, Klecka & Konvicka*, *2011*).

**Table 1** Comparative response occurrence of females Eurasian sparrowhawk (*Accipiter nisus*) when incubating eggs and brooding nestling in the Czech Republic in 2013.

| | Incubation | | | | | Brooding | | | |
|---|---|---|---|---|---|---|---|---|---|
| Locality/response | Skittish | Alert | Angry | Intense | Locality/response | Skittish | Alert | Angry | Intense |
| Urban | 27% | 0% | 21% | 53% | Urban | 21% | 0% | 16% | 63% |
| Rural | 47% | 18% | 29% | 6% | Rural | 18% | 35% | 29% | 18% |

**Table 2** Data analyses of the responses of females Eurasian sparrowhawk (*Accipiter nisus*) to the human approach in the Czech Republic in 2013—effect of variables on the female's response (response 1, skittish; response 2, alert; response 3, angry; response 4, intense).

| | Reaction 1 | | | | Reaction 2 | | | | Reaction 3 | | | | Reaction 4 | | | |
|---|---|---|---|---|---|---|---|---|---|---|---|---|---|---|---|---|
| | $\chi^2_1$ | $P$ | Est. | SE | $\chi^2_1$ | $P$ | Est. | SE | $\chi^2_1$ | $P$ | Est. | SE | $\chi^2_1$ | $P$ | Est. | SE |
| Locality: stage | 2.79 | 0.09 | 0.24 | 0.15 | 3.85 | **0.04** | 0.24 | 0.15 | 0.09 | 0.75 | −0.52 | 0.17 | 0.01 | 0.90 | −0.01 | 0.11 |
| Stage | 4.80 | **0.02** | −0.29 | 0.11 | 3.13 | 0.07 | −0.29 | 0.11 | 0.11 | 0.73 | 0.00 | 0.12 | 4.24 | **0.03** | 0.12 | 0.08 |
| Locality | 0.01 | 0.90 | −0.11 | 0.15 | 6.07 | **0.01** | −0.11 | 0.15 | 0.79 | 0.37 | −0.08 | 0.02 | 6.32 | **0.01** | 0.35 | 0.15 |
| Path | 0.42 | 0.51 | 0.04 | 0.06 | 0.76 | 0.38 | 0.04 | 0.06 | 0.22 | 0.63 | 0.03 | 0.06 | 2.90 | 0.08 | −0.11 | 0.06 |
| Habitat | 3.30 | 0.06 | −0.17 | 0.1 | 0.16 | 0.68 | −0.17 | 0.35 | 0.32 | 0.57 | 0.05 | 0.09 | 0.96 | 0.32 | 0.09 | 0.10 |

Frequency of each reaction type during incubation and brooding was proportionally expressed as a result of presence of the reaction from the total of observations.

## RESULTS

The responses of females were observed at 17 rural nest sites and 19 urban nest sites. All four types of nest defense were observed in both rural and urban environments. The most frequent behaviors observed were the extreme reactions, i.e., either the first (skittish) or the fourth (intense) types, whilst the second (alert) type of behavior was relatively scarce (Table 1).

The average distance from the nest to the nearest trail was 65.0 m (±59.94 SD) in the rural area and 33.6 m (±29.95 SD) in the urban area. The distance from the path to the nest and the habitat type did not affect the females' response to human approach. The locality played a significant role in the "alert" ($\chi^2 = 6.07$, $Df = 1$, $P = 0.01$) and the "intense" ($\chi^2 = 6.32$, $Df = 1$, $P = 0.01$) responses, with "alert" being most commonly observed at rural sites, while the "intense" response was most common at urban nests. Reluctance to leave the nest and aggressive behavior towards the human, classified as "intense," were more frequent at urban sites (84.6%) than at rural sites (15.4%). Although not statistically significant, a trend was found in the effect of stage of breeding on the female's behavior in the "skittish" ($\chi^2 = 4.80$, $Df = 1$, $P = 0.02$) and "intense" ($\chi^2 = 4.24$, $Df = 1$, $P = 0.03$; Table 2.) responses. The first type response; females left the nest more willingly while incubating eggs (65.0%) than while brooding young (35.0%). The fourth type response; females refused to leave the nestlings more than when incubating (57.7% vs. 42.3% resp.). The third type of behavior was not affected by any of the explanatory variables.

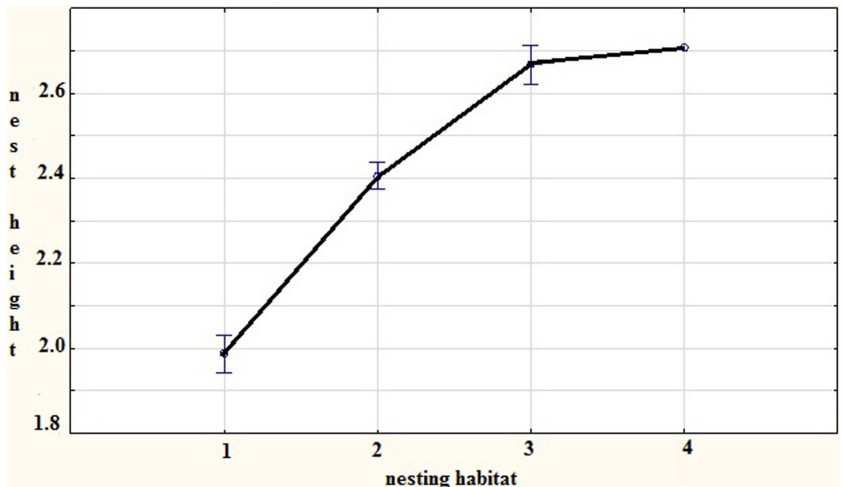

**Figure 1** The relation of nest height (log transformed) on vegetal density in close vicinity of the nest tree—habitat type 1 is dense and that impedes access, habitat 2 and 3 are intermediate in density to 1 and 4 and habitat type 4 is sparse, Czech Republic, 2013.

We found that the height at which the nest is built is dependent on the density of the vegetation surrounding, the nest tree ($F = 34.182$, $Df = 3$, $P < 0.001$). The sparrowhawks placed their nests lower in trees with denser vegetation (Fig. 1).

## DISCUSSION

Habituation of wildlife to human settlements has been noted since humans first created collective communities. However, the sensitivities and the degree to which each species responds are not yet understood and probably changes on an annual or generational basis. Many domesticated animals, especially dogs (*Canis familiaris*), are known to follow the human gaze and understand what is required of them, or aim to please their human companion, (*Horowitz, 2010*). In addition, the study of animal behavior is also known to greatly contribute to conservation of the studied species in the wild (*Clemens & Buchholz, 1997*). Hence, we examined nest defense behavior of female sparrowhawks during the incubation and the brooding stages of the reproductive cycle, as a function of distance from the human-frequented trail to the nest and rural/urban habitats. Our hypothesis was repudiated because habituation to humans had a negative influence on the sparrowhawks and their reactions were significantly more aggressive in the urban area. Even though sparrowhawks in Prague are exposed to nonthreatening humans more often than are sparrowhawks in rural areas, and have greater opportunities to learn from their interactions, their responses were more aggressive. Such aggressiveness in urban breeding birds was observed also in other species (*Knight, Grout & Temple, 1987*). This does not support our hypotheses wherein we thought that the continuous exposure and contact with humans would allow the sparrowhawks to habituate to the constant disturbance. On the contrary, we discovered that the females in urban areas appeared to have much higher stress levels than those in the rural areas. In rural areas, breeding sparrowhawks avoided humans in spite of minimal or no prior contact. In Japan, *Abe et al. (2007)* demonstrated that in

spite of the absence of persecution, sparrowhawks chose to nest in relatively undisturbed portions of habitat, in comparison to places of increased human activity.

Humans are known to impact wildlife with almost every activity in which we indulge (*Glasson, Godfrey & Goodey*, *1995*; *Knight & Gutzweiller*, *1995*; *Rein & Scharpf*, *1997*). The distance from the path to the nest was shorter in Prague where parks are regularly used by humans for recreational purposes. Our findings concur with *Smith, Bosakowski & Devine* (*1999*) who also found a similar pattern in other avian species nesting in both rural and urban environments. The recreational path represents an open space for flying birds and adults often take advantage of them especially when bringing food to the nest (*Kenward*, *2006*). Besides the fact that it is easily assumed that nesting birds in urban areas should be habituated to human presence, the aggressive reaction was greatest in these surroundings. This leads us to consider the point that we assume that urban wildlife undergo a habituation process and hence are able to live in urban areas. In a high disturbance environment habituation to the disturbing stimulus is assumed to be adaptive, and in urban settings this is likely to take the form of noise or humans passing close to the nest. These will be largely harmless, so in this situation, habituation would surely be of benefit to the birds, as they will spend more time at the nest and also draw less attention to the location of the nest site. However, the extreme behaviors displayed by the female sparrowhawks in the defense of their nests actually suggest this widespread assumption to be incorrect and that wildlife, regardless of the amount of human presence, does not really adjust to our activities. In fact, wildlife remains apprehensive of our presence and the high levels of disturbance appear to result in increased levels of aggression towards those that disturb the nest-tree, irrespective of the "habituation."

Our findings suggest that the distance from the path to the nest is not as important for choosing the nest site as the structure of the surrounding woods and the specific tree chosen in which to build the nest. These findings are similar to those of *Newton* (*1986*). Furthermore, it appears that although the sparrowhawks have adapted to living in urban areas they have not adapted to human disturbance, nor lost their defensive mechanisms. Our results suggest that the urban breeding pairs are constantly at greater stress levels than conspecifics breeding in rural areas. In consideration of the significant fact that raptors breed in urban environments (e.g., *Bird, Varland & Negro*, *1996*; *Berry, Bock & Haire*, *1998*), it is important to elucidate the physiological differences between the two populations in future studies, by evaluating the different stress levels using diverse techniques such as corticosterone levels (e.g., *Bortolotti et al.*, *2008*; *Bortolotti et al.*, *2009*) or ptilochronology (e.g., *Gombobaatar, Yosef & Odkhuu*, *2009*).

One of the main predictors of the female sparrowhawks' response to human approach was the stage of the breeding cycle. In both, the urban and rural sparrowhawks, the females defended their broods more tenaciously than when incubating their eggs. The later the provocation was in the breeding cycle, the more aggressive the reaction of the birds was to humans, and/or the more reluctant the female sparrowhawks were to leave the nest. Similar results were attained in other avian species that were also approached by humans (*Andersen*, *1990*; *Sproat & Ritchinson*, *1993*; *Sergio & Bogliani*, *2001*) supporting the parental investment theory (*Trivers*, *1972*).

Two other factors that were not evaluated in our study but could potentially have an impact on a females' behavior are the age and breeding experience of the female and clutch size. If the suggestions of *Knight & Temple* (*1986*) are correct then it is possible that older females, with increasing experience, will react differentially to humans as compared to younger, inexperienced females. Also, in order to further verify the parental investment theory (*Trivers*, *1972*) we consider it probable that females will defend larger clutches more vigorously as compared to smaller ones, despite the study of *Osiejuk & Kuczynski* (*2007*) who found the effect of clutch size on flushing distance was small and did not support the parental investment theory. In general, the sparrowhawk's behavior could also be altered by human persecution. This study does not reveal whether the reduced nest defense in rural sparrowhawks is a change in gene frequency due to persecution of more aggressive and less cautious individuals or whether increased aggressiveness in urban sparrowhawks is learned as a result of the greater frequency of disturbance.

Another important point that resulted from this study is the fact that sparrowhawks display the ability to choose the height at which to place their nest as a function of the density of the vegetation surrounding the nest tree (Fig. 1). We assume that this may not really be the case and that sparrowhawks are unable to discern or evaluate vegetation density, but recognize niches that will hinder the approach of a potential predator. We think that because flight is the most expensive behavior (cf. *Norberg*, *1995*), especially when carrying extra weight in the form of prey to the nest; it would be advantageous for the breeding pair to nest low and to save the energy invested in flying to the higher parts of the tree. This concurs with *Newton* (*1986*), who also found that nest height was dictated by vegetal density and age.

## CONCLUSIONS

We found that female sparrowhawks breeding in the urban environment of the city of Prague displayed a greater degree of aggression and agitation than their conspecifics that breed in rural areas. Furthermore, we found that parental investment in the form of nest defense increases with the breeding cycle and that parents with nestlings showed greater levels of defense behavior compared to when they were incubating eggs. In addition, in order to better defend their nests from predators, and yet to enable easier access to them, it appears that in dense vegetation, nests were placed comparatively lower than those in more open vegetation.

## ACKNOWLEDGEMENTS

We thank Miroslav Šálek and David Horobin for comments on an earlier version of the paper and Sue Har-Shefi for the improvement of the language.

### Funding

This project was funded by the Internal Grant Agency of the Faculty of Environment, Czech University of Life Sciences, Prague (grant no.: 20134251 and 20144233). The funders had

no role in study design, data collection and analysis, decision to publish, or preparation of the manuscript.

## Grant Disclosures
The following grant information was disclosed by the authors:
Internal Grant Agency of the Faculty of Environment, Czech University of Life Sciences, Prague: 20134251, 20144233.

## Competing Interests
The authors declare there are no competing interests.

## Author Contributions
- Tomas Kunca conceived and designed the experiments, performed the experiments, analyzed the data, contributed reagents/materials/analysis tools, wrote the paper, prepared figures and/or tables, reviewed drafts of the paper.
- Reuven Yosef contributed reagents/materials/analysis tools, wrote the paper, reviewed drafts of the paper.

## Animal Ethics
The following information was supplied relating to ethical approvals (i.e., approving body and any reference numbers):

The author was authorized and certified to conduct experiments/research on vertebrate animals according to the act no. 246/1992 Coll. on protection of animals against cruelty.

The animals were not handled during the experiment; therefore there is no need for approval.

## Data Availability
The raw data has been supplied as Data S1.

## Supplemental Information
Supplemental information for this article can be found online at http://dx.doi.org/10.7717/peerj.2070#supplemental-information.

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
