# Peer review of "Differential nest-defense to perceived danger in urban and rural areas by female Eurasian sparrowhawk (Accipiter nisus)"

_PeerJ, doi:10.7717/peerj.2070_

## Round 0.1 · original submission · Major Revisions

Dear authors

As you can see our reviewers made a number of suggestions to improve your ms. If you are willing to revise we would be happy to review the revision again.

Kind regards

Michael Wink
Academic editor

·

Basic reporting

Writing in the subject manuscript is generally OK, but there are grammatical and typographical errors throughout. For example, on line 99, the sentence includes the phrase “data was.” Data is a plural word so it should read “data were.” Also, on line 133 (and in several other places in the manuscript), the word “using” is a dangling participle in that it modifies “nesting tree” (not what the authors intended!). The best way to correct this error is to write in first person as follows: We used a Nikon Forestry PRO to measure nest height and distance….. Most of the grammatical and typographical errors can easily be corrected, but will require a careful read.

I noted that in the abstract and elsewhere the authors used the term “suitable habitat.” Habitat, by definition, is an area that has the resources and environmental conditions needed to support members of a given species, thus suitable habitat is redundant. The authors could use the term suitable environment or just habitat.

The introduction does a good job of setting the stage for the study, and pertinent references were included. The authors appeared to follow PeerJ Standards and raw data were supplied.

Both the figure and table presented should be modified. In Figure 1, all habitat types should be described. I am assuming that habitat types 2 and 3 are intermediate in density to 1 and 4, but that is not clear. In Table 1, the responses should be identified and the data should be presented in a way that the numbers can be compared (i.e., percentages as opposed to raw values). In Table 2, the responses should be identified. In all figures and tables, the titles or legends should include place and dates of the study.

Experimental design

The subject matter of the manuscript appears to fit within the scope of the journal, and fills an informational gap. I think most biologist who have studied Accipiters recognize that most are fearful of humans in non-urban settings, but are much less so in urban areas. But those differences in behavior have not been quantified (to my knowledge), so the manuscript makes a contribution.

The methods used in the study are simple, but adequate to answer the question posed.

Validity of the findings

Data presented are clear and unambiguous, and the conclusions drawn are sound. The authors framed their study under the assumption that habituation to humans would result in less aggressive behavior towards people. Another way to look at the idea of habituation is that as Accipiters become habituated to humans they lose their fear of them and then treat them like any other potential predator (i.e., aggressive actions as opposed to flying away).

Additional comments

I enjoyed reading the manuscript, and it adds to our understanding of animal behavior in human dominate landscapes.

·

Basic reporting

I had trouble in a few places following the train of thought, and believe the article could be improved by another round of careful editing.

Experimental design

I was confused by what seems to be a disconnect between the Methods and Results. In the Methods, you described essentially a random-effects ANOVA design that makes sense conceptually, but in the Results you use and present counts of reactions by response level and compare test them using a Chi-square test. This doesn’t match what I was expecting from the Methods section, but does seem a more appropriate approach given the categorical nature of the response variable. I suggest you re-frame the Methods to focus on a simple test of the intensity of response using a Chi-square. Alternatively, you could recast your responses as ordinal (with the intensity of the response increasing sequentially from level 1 to 4), and use an ANOVA with model selection as you describe in the Methods section. However, a challenge with this approach is that most of your responses were at the extremes, thus you are not dealing with a normally distributed response in either setting and you would need to adopt a more complex model to account for that.

Validity of the findings

The results appear to be fairly compelling with respect to a difference in response intensity between rural and urban sparrowhawks, thus I believe there are valid findings to report.

Additional comments

I enjoyed reading the paper and appreciate the opportunity to provide comments. I was surprised to not find a response level indicating the female attacked the human intruder. Did that not occur? I work with Cooper's hawks, and such attacks, including hawks striking humans, are fairly common in my urban study area.

---

## Round 0.2 · accepted · Accept

Dear authors

Good news- your revision is accepted and your paper will be published soon. Thanks for submitting your work to PeerJ.

Regards
Michael Wink